# Gait Analyses of Parkinson's Disease Patients Using Multiscale Entropy

**Yuan-Lun Hsieh [1] and Maysam F. Abbod [2,\*]**

1 Department of Physical Therapy and Assistive Technology, National Yang Ming Chiao Tung University, Taipei 11221, Taiwan; stephanie8787@gmail.com
2 Department of Electronics and Electrical Engineering, Brunel University London, Uxbridge UB8 3PH, UK
\* Correspondence: maysam.abbod@brunel.ac.uk

**Abstract:** Parkinson's disease (PD) is a type of neurodegenerative diseases. PD influences gait in many aspects: reduced gait speed and step length, increased axial rigidity, and impaired rhythmicity. Gait-related data used in this study are from PhysioNet. Twenty-one PD patients and five healthy controls (CO) were sorted into four groups: PD without task (PDw), PD with dual task (PDd), control without task (COw), and control with dual task (COd). Since dual task actions are attention demanding, either gait or cognitive function may be affected. To quantify the used walking data, eight pressure sensors installed in each insole are used to measure the vertical ground reaction force. Thus, quantitative measurement analysis is performed utilizing multiscale entropy (MSE) and complexity index (CI) to analyze and differentiate between the ground reaction force of the four different groups. Results show that the CI of patients with PD is higher than that of CO and 11 of the sensor signals are statistically significant ($p < 0.05$). The COd group has larger CI values at the beginning ($p = 0.021$) but they get lower at the end of the test ($p = 0.000$) compared to that in the COw group. The end-of-test CI for the PDw group is lower in one of the feet sensor signals, and in the right total ground reaction force compared to the PDd group counterparts. In conclusion, when people start to adjust their gait due to pathology or stress, CI may increase first and reach a peak, but it decreases afterward when stress or pathology is further increased.

**Keywords:** multiscale entropy; Parkinson's disease; gait impairment; dual task; complexity index

## 1. Introduction

The four cardinal signs of Parkinson's disease (PD) are tremor, bradykinesia, rigidity, and postural instability [1]. These symptoms can influence patients' gait pattern and may lead to gait impairments. For PD patients, alterations in walking, including reduced gait speed, shortened stride, reduced swing times, and decreased arm swing, are common [2].

When performing dual tasks, people rely upon executive function and the ability to divide attention. Neuroimaging indicated that a dual task highlights the role of a higher level of cognitive and frontal lobe function. However, if the gait and secondary tasks demand attention, performance of at least one of the tasks will deteriorate. Unfortunately, the cognitive functions of patients with PD are generally impaired [2]. One meta-analysis found that the negative effect of dual tasks is present regardless of the mean level of single-task gait speed in a study. In addition, dual task walking speed deteriorates regardless of the type of dual task [3].

To analyze the patients' gait, two phases need to be defined: stance phase, in which the foot contacts the ground, and a swing phase, in which the foot advances in the air. Normally, the stance phase accounts for 60% of the gait cycle, while the proportion of the swing phase is 40% [4]. Clinically, gait analysis is often done by observation. Quantitative measurements are rarely used. For instance, the Timed Up and Go Test is used commonly. Although the Timed Up and Go Test may give us the information of gait speed, it is hard to relate speed to gait patterns and quality [5]. Other than health professionals, it is difficult for

lay people to observe an imbalance between two sides or abnormal timing when entering stance or swing phase.

Due to the reasons mentioned above, a simple and clear quantitative measurement is required to document gait variability in PD patients. Applying a linear method such as Time Up and Go may not be that appropriate due to the gait signals being nonlinear. When people walk, the fluctuations between strides, gait speed, total reaction force, and swing intervals can all lead to gait variability. For healthy people, the gait variability seems to follow certain temporal structures. On the other hand, for PD patients, their fluctuations become completely random. This process of increasing the system's order until forming a stationary state is called self-organization, and this cannot be analyzed through means of closed system [6]. As a result, it is important to find a method for analyzing nonlinear signals. As known, entropy is a concept that addresses system randomness and predictability [7]. A variety of measures with the concept of entropy have been proposed, including approximate entropy, sample entropy, corrected conditional entropy, fuzzy entropy, compression entropy, permutation entropy, distribution entropy, multiscale entropy, self-entropy, and information storage [8]. People like to use entropy measures due to their convenience to analyze the dynamic activity of real-world systems.

However, Xiong et al. [8] mentioned that stationarity in time series is a prerequisite when applying entropy measures. In addition, traditional entropy-based algorithms, such as approximate entropy and sample entropy (SE), are not always associated with dynamical complexity for estimating entropy [9]. Furthermore, entropy estimates are often highly dependent on the method-specific parameters, hence different results are needed after changing measures. Therefore, advanced methods such as MSE would be appropriate and reveal changes of complexity during walking. Of these methods, multiscale entropy (MSE), which is an extension of SE, had been used broadly for analyzing EEG changes, heart rate variability, brain consciousness, and even the signals of an electromyogram [10]. It measures the structural richness of information over multiple temporal scales. Because traditional entropy measures quantify only the regularity of time series on a single scale, MSE can measure the complexity of the system [11–15]. However, there is no direct relation between regularity and complexity. Originally, complexity is associated with meaningful structural richness incorporating correlations over multiple spatiotemporal scales. That is why the calculation of the area under the sample entropy for different scales is done using MSE [16]. MSE has been successfully utilized to analyze several bio-signals and to distinguish healthy status from pathological conditions. A higher entropy value reflects an increase in the degree of randomness but not necessarily an increase in the complexity of the time series. For instance, white noise series have high entropy but in fact low complexity [11]. With data about the ground reaction force, the complexity from MSE of healthy controls (CO) and PD patients can be calculated. Furthermore, we can estimate the complexity of gait from different groups.

To date, although there are studies using MSE, this is the first paper to identify gait impairment of PD patients using MSE in order to identify the pathological influences. Next, the complexity differences between PD and CO are compared. Lastly, we investigate how further stress influences the MSE between subjects performing with and without dual tasks.

## 2. Materials and Methods

### 2.1. Data Acquisition and Analysis

The data used in this study were retrieved from PhysioNet [17]. The study included 93 patients with idiopathic PD (mean age: 66.3 years; 63% men), and 73 healthy controls (mean age: 66.3 years; 55% men). Since this study is aimed at comparing groups with dual tasks, only 21 PD patients and 5 healthy controls were included. The patients were classified using the Hoehn and Yahr Scale [18] as follows: 12 people in stage 2, 6 people in stage 2.5, and 3 people in stage 3. Higher stages indicate having more serious symptoms.

The sensors used to measure vertical ground reaction force were placed inside a shoe insole with eight sensors on each foot, as shown in Figure 1. The sensors were made by Ultraflex Computer Dyno Graphy, Infotronic Inc. [19], and they recorded changes in force at a sampling rate of 100 Hz [20]. The highest frequency, feasible for analysis was set by the sampling rate at which the data were collected. However, MSE analyses at 10 Hz frequency should be performed on data with at least 50 Hz sampling rate [21]. In this study, the estimated frequency rate for walking was well under 10 Hz, hence a sampling rate of 100 Hz was adequate. In addition, the recording unit was carried at the waist. By using the pressure sensors, the average stride time, swing time (%), stride time variability, and swing time variability can be measured, this method is based on work presented by Yogev et al., 2005 [2]. Subjects were instructed to walk at their usual, self-selected pace for approximately 2 min on level ground. While for the dual-task group, subjects walked while doing serial 7 subtraction.

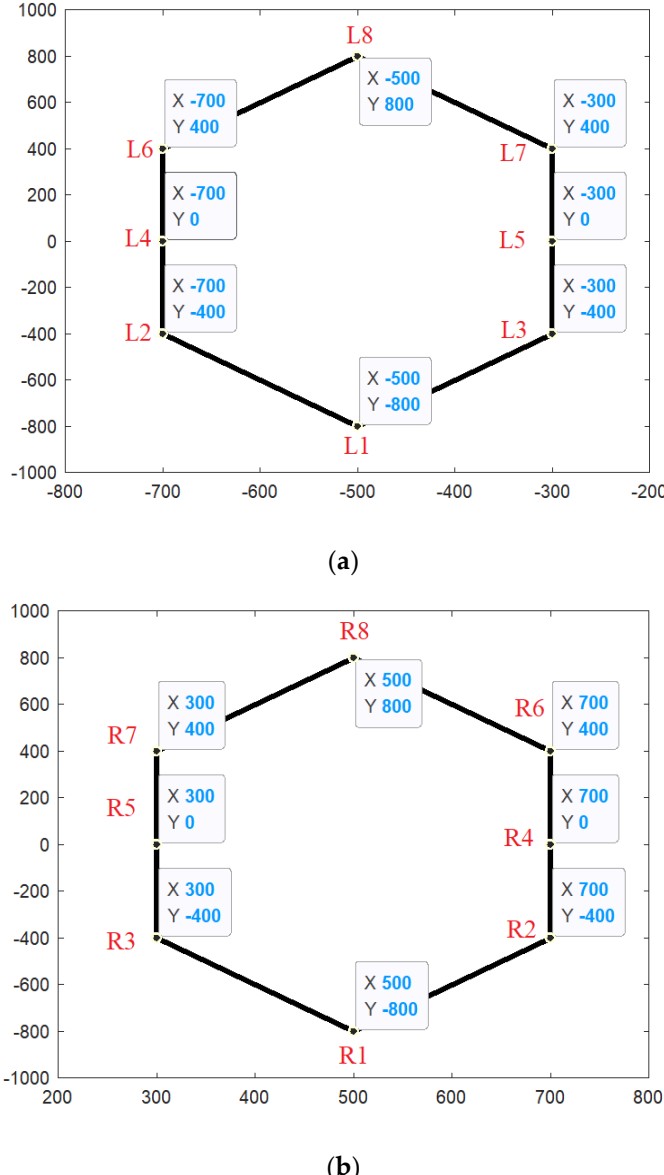

**Figure 1.** Locations of eight sensors: (**a**) left foot and (**b**) right foot.

After analyzing the basic gait data, such as gait speed and stride time, the idea of using entropy to further analyze ground reaction force data was proposed. The reason for

choosing MSE is supported by previous work that has shown MSE to be superior to SE when analyzing bio-signals [16,22].

When analyzing the data, the 2 min was cut into two parts. The first part (T1) started from 1 to 20 s and the second part (T2) was from 21 to 120 s. That is because the first 20 s was the time taken for the subjects to adjust themselves to fit the task. The two periods were selected such that a comparison between the adaptation of CO and PD could be made. Furthermore, adaptation may be important afterward when the pressure from pathology or the dual task overwhelmed them. When the adaptation is finished and the subjects start walking normally, it will be easier to predict patterns when facing obstacles. For the first 20 s, there were fewer signal changes, so the scale was set to 10 due to the MSE calculation's total number limitation. However, for the second part, the time span was longer, so the scale was set to 20.

### 2.2. Sample Entropy and Multiscale Entropy Analysis

For gait analysis, it is important to know how SE and MSE differ among these groups. Based on SE, MSE is a method to evaluate the complexity of signals over different time-scales [22].

SE, as shown in Equation (1), depends on three important parameters that have been fully researched and proved by Costa et al., 2002 [16]. Including the length of the epoch ($N$), the number of previous values used for the prediction of the subsequent value ($m$), and a filtering level ($\gamma$). The $\gamma$ parameter is usually set as a percentage of the standard deviation (SD) of the time series [23]. If a vector of length $N$ has repeated itself in tolerance $\gamma$ for m points, it will also do so for the $m + 1$ point. $Sm(\gamma)$ is the average amount of Si m($\gamma$) for $i \in [1, N - m]$, and $Sm + 1(\gamma)$ is the average of $m + 1$ consecutive points. In this study, T1 was set to $N = 2000$, and for T2 it was $N = 10,000$. This is due to setting the sampling rate at 100 Hz, so for T1 (20 s) there were 2000 data points and 10,000 data points for T2 (100 s). The default values of the parameters were $m = 2$ and $\gamma = 0.15$ [16]; however, different settings were tested, and it was concluded that $m = 2$ and $\gamma = 0.1$ gives the best results. Moreover, by the nature of the calculations, SE for periodic, regular signals is approximately zero while it is maximal with irregular, random signals. However, accurate entropy calculation requires a vast amount of data, and the results will be affected by system artifacts and noise. Hence, it is not practical to apply entropy methods to real applications. Approximate entropy and SE were developed to handle these limitations to be able to handle small data problems. It has been suggested that 200 data points per window are needed to obtain consistent SE values [23], though many studies have used 600 data points at the longest time scale [21,24,25]. In addition, according to Richman and Moorman [26], the ranges of $N$, $m$, and $\gamma$ have been analyzed for sets of random numbers with known probabilistic character to compare with approximate entropy.

$$\text{SE}(N, m, \gamma) = -\ln \frac{S^{m+1}(\gamma)}{S^m(\gamma)} = -\ln \left[ \frac{(N-m-1)^{-1} \sum_{i=1}^{N-m-1} S_i^{m+1}(\gamma)}{(N-m)^{-1} \sum_{i=1}^{N-m} S_i^m(\gamma)} \right] \qquad (1)$$

MSE has a coarse-grain procedure for the data vector $X_N = \{x_1, x_2, \ldots, x_N\}$ before SE computation, which is the main computation difference between MSE and SE. MSE can measure the distribution of complexity on multiple time scales. At the first scale, the MSE algorithm evaluates SE for the time series at each sampled point. Then, SE is computed on coarse-grained versions of the original time series in greater scales. $y$ is a set of data divided into nonoverlapping frames of length $\tau$, where $\tau$ represents the scale factor and takes integer values equal to or greater than 1, which can be found using Equation (2). Therefore, $\tau$ is defined by length of $N$ of $X_N$ and $m$. For example, $N = 10,000$ and $m = 2$, take 500 as the median of $[10^m, 30^m]$ [27], which is selected to calculate $\tau_{\max} = 10,000/500 = 20$ and $\tau \in [1,20]$. By using different scales, further data analysis can be made and more information about the gait patterns can be extracted. However, attention should be paid when setting a constant r, as more and more patterns will be considered indistinguishable

while increasing $\tau$, thus artificially decreasing the entropy rate and increasing regularity with the scale factor [28,29].

$$y_j^{(\tau)} = \frac{1}{\tau} \sum_{i=(j-1)\tau+1}^{j\tau} x_i \qquad (2)$$

Then, the $SE_\tau(N, m, \gamma, \tau)$ equation can be calculated by $y^{(z)}$ using Equation (2). Once the time scales of the interest range have been identified, the area under the $SE_\tau(N, m, \gamma, \tau)$ versus time-scale $\tau$ curve, known as the complexity index (CI) [30] can be calculated using Equation (3).

$$CI = \sum_{1}^{\tau_{max}} SE_\tau(N, m, \gamma, \tau) \qquad (3)$$

Figure 2 presents how to test the gait time series models for stationarity from Unit Root test in MATLAB [31]. A previous paper [8] mentioned that stationarity of the time series is a prerequisite when applying entropy measures. Figure 2a is a standard graph showing test simulated data to compare with other graphs and see whether the signals showed signs of stationarity. The differences between the series are clearer here—the trend of the stationary series has little deviation from its mean trend. While the different stationary and heteroscedastic series have persistent deviations away from the trend line. Last, the ARIMA (AR(1)) model series exhibits long-run stationary behavior, which is more similar to our CO and PD raw data of feet force. Hence, the augmented Dickey–Fuller test was used on AR(1) series to assess whether the series had a unit root. The result was true, so it is a stationary series. However, when using the KPSS test to assess whether the series was unit root nonstationary, the result was still true, so it was nonstationary. This means that the AR(1) series was a mix between stationary and nonstationary signals. Interestingly, it was found that the raw data (i.e., all 5 CO subjects and randomly selected 10 PD from 21 PD patients) were tested by Augmented Dickey–Fuller and KPSS tests. The four CO subjects and eight PD patients exhibited a stationary signal, but one CO and two PD patients were similar to AR(1), which was a mixed stationary and nonstationary signal as shown in Figure 2b,c, which depicts one example from CO and PD presenting raw data of the ground reaction force. This was not surprising for these results because CO subjects and PD patients have sometimes shown their gait pattern to be stationary or nonstationary. That is why MSE is used to analyze these raw data, for analyzing which, it would not be suitable to only utilize entropy measures such as sample entropy.

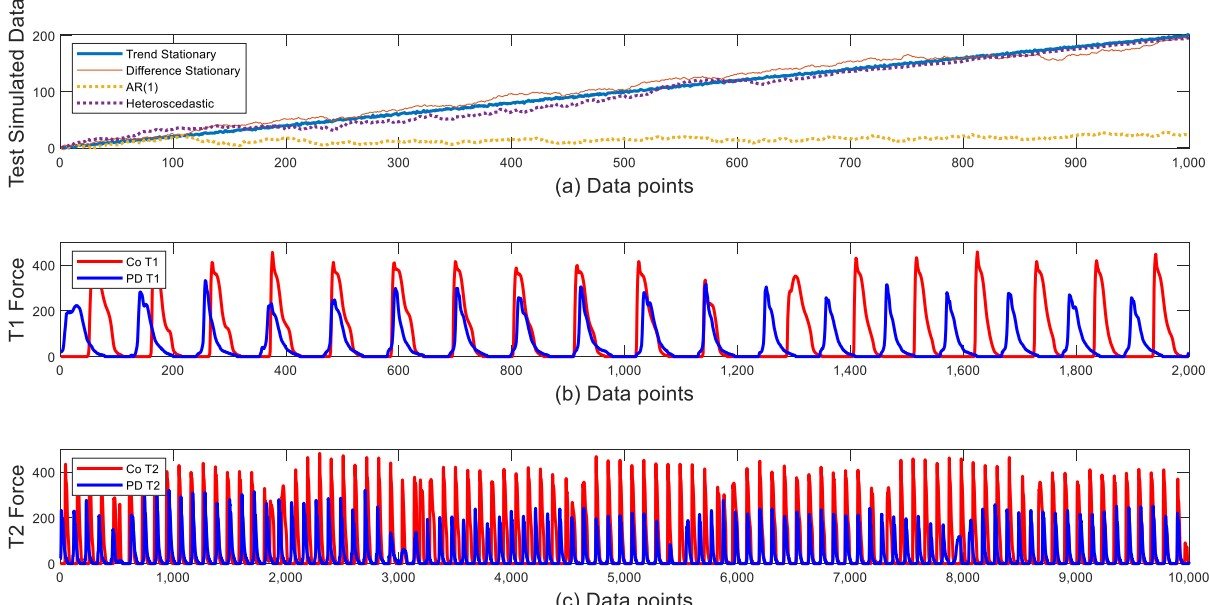

**Figure 2.** Test simulated data versus T1/T2 ground reaction force signals (**a**) Test simulated data; (**b**) T1 ground reaction force signals; (**c**) T2 ground reaction force signals.

*2.3. Statistical Analysis*

To compare the significant difference of demographics, the gait speed and CI of CO and PD groups were compared using sample *t*-test. For the normality of the distributions, Levene's test of equal variances was applied first to test whether the two groups of variances were equal or not. Then, following the results of the variances, two different *p* values were obtained. Additionally, to identify the changes with and without dual tasks, paired sample *t*-tests were utilized. IBM SPSS Statistics 24 software was used for performing the *t*-test analysis. The *p* value was considered statistically significant if it was less than 0.05.

## 3. Results

The demographic data of 21 PD patients and 5 healthy controls details are mentioned in Table 1. Age, weight, BMI and gait speed showed significant difference. From the table, it can be seen that the proportion of males are higher in PD group.

**Table 1.** Demographic data of the subjects.

| | CO (*n* = 5) | PD (*n* = 21) | *p* Value |
|---|---|---|---|
| Age (years) | 66.000 ± 2.000 | 71.860 ± 7.316 | 0.004 * |
| Sex (Male %) | 40% | 81% | 0.068 |
| Hoehn–Yahr Scale | NA | 2.286 ± 0.373 | NA |
| Height (m) [1] | 1.660 ± 0.07 | 1.720 ± 0.111 | 0.239 |
| Weight (kg) [2] | 57.000 ± 8.602 | 78.200 ± 14.6672 | 0.002 * |
| BMI (kg/m$^2$) [3] | 20.860 ± 3.725 | 26.794 ± 3.6453 | 0.005 * |
| Gait speed (m/s) | 1.389 ± 0.095 | 1.001 ± 0.256 | 0.003 * |
| Gait speed: dual (m/s) [4] | 1.305 ± 0.140 | 0.926 ± 0.1534 | 0.001 * |
| Stride time (s) | 8.506 ± 1.142 | 7.134 ± 3.001 | 0.331 |
| Stride time: dual (s) | 8.608 ± 5.393 | 6.629 ± 3.128 | 0.281 |

Data are expressed as mean ± SD. CO: healthy controls, PD: Parkinson's disease. Hoehn–Yahr Scale [10]: 12 in stage 2, 6 in stage 2.5, and 3 in stage 3, * *p* < 0.05. [1] *n* = 18 (no data available for PD15, 19, 21), [2] *n* = 20 (no data available for PD23), [3] *n* = 17 (no data available for PD15, 19, 21, 23), [4] *n* = 7 (data available only for seven patients: PD13–PD19).

Figure 3 is an example of real-time CI values for CO and PD patients obtained from all 16 feet sensors and the calculated total ground reaction force. It can be seen from Figure 3a–d, there are subtle differences between each sensor on the graph and the MSE values. Figure 3a,c looks similar with rising slope. However, CO and PD data show a different appearance at the end in Figure 3b,d, with more changes in PD, which is getting to a peak and then decreasing to a low value. Neither of the sensors had same values nor did the curvature of the graph look the same. Furthermore, even for the same person in the CO or PD groups, both sides demonstrated different characteristics. This example can give a quick overview of the data for each subject.

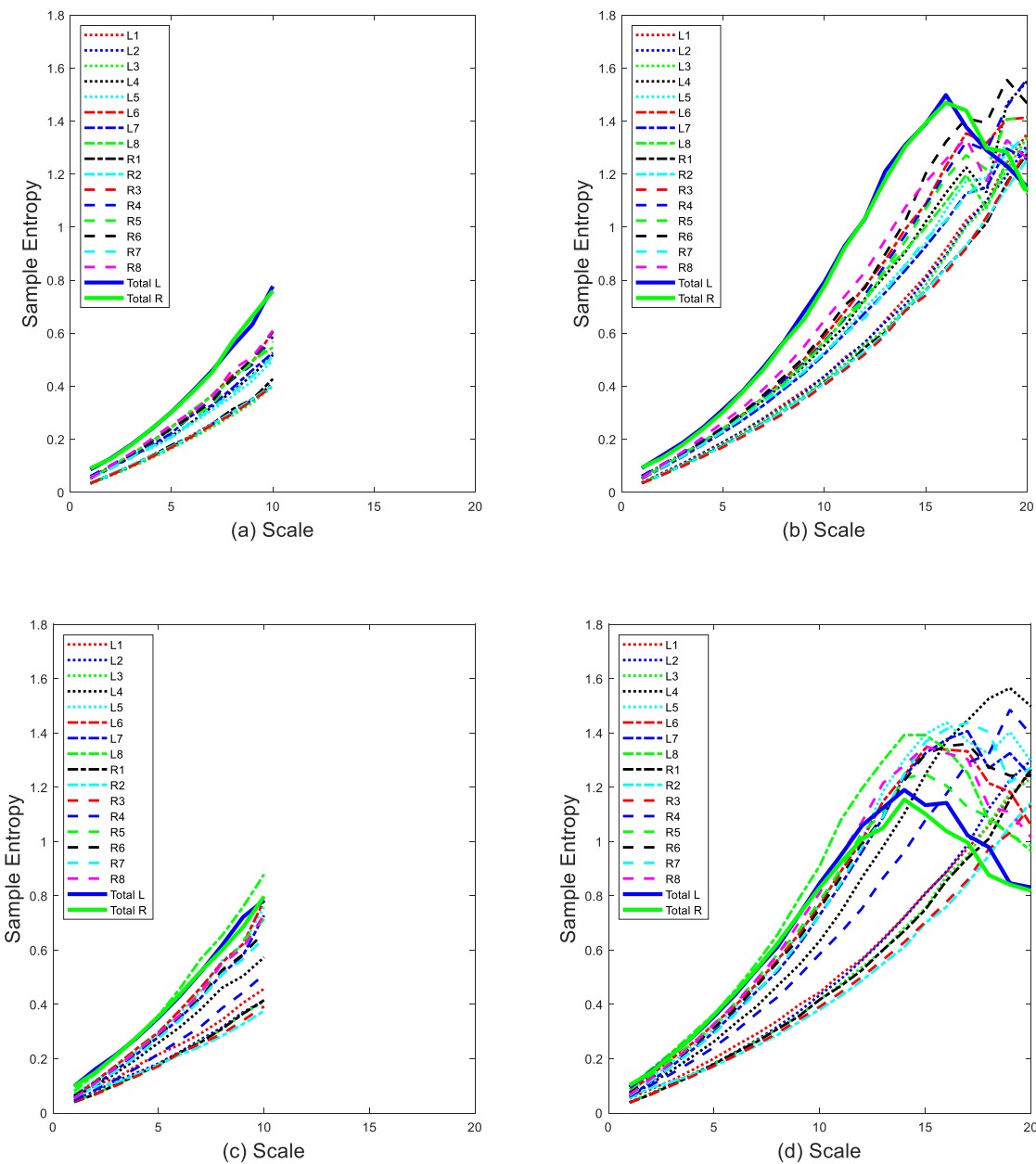

**Figure 3.** One example of MSE on 16 sensors (**a**) CO: T1; (**b**) CO: T2; (**c**) PD: T1; (**d**) PD: T2.

### 3.1. Comparison of SE and CI for CO and PD

Table 2 shows the differences between SE and CI of CO and PD. The data indicate that PD had larger SE and CI values except for the total ground reaction force. Five left sensors (L1, 2, 3, 6, 7) and six right sensors (R1, 2, 3, 6, 7, 8) showed statistical significance in T1 CI data; seven left sensors (L1 to 4, 6–8) and four right sensors (R 2, 6–8) also presented similar results in T2 data. Additionally, the SE of 10 sensors showed significant differences in both T1 and T2 phases. Although SE may seem to be a rather simple way of analysis, it is important to indicate that the ground reaction force data contain stationary and nonstationary signals, hence using SE alone will not be accurate. Here is an example showing the differences when using SE and CI. Figures 4 and 5 show examples of the L2 and R2 position of T1 and T2 for 21 PD and 5 CO subjects' MSE. From Figure 4a,b, with reference to the results in Table 2, the PD patients had larger values. In addition, similar results are shown in Figure 3c,d. In Figure 4b,d, the slope of all the curves is positive and the curves started close together but at the end they diverged into several

lines spanning different values of SE. Figure 5 shows the results of the T2 phase where PD had larger values than CO. Observing Figure 5a,c, the curves of L2 and R2 sensors showed similar results. On the other hand, Figure 5b,d seems to be varied and had more wavy lines toward the end. Moreover, the box plot in Figure 6 shows L2 and R2 positions, in which significant differences between the two groups were clearly demonstrated in the positions at T1 and T2. However, the PD group showed a larger standard deviation.

**Table 2.** SE and CI of T1 and T2 for CO and PD.

|  | T1 CO (*n* = 5) | T1 PD (*n* = 21) | T2 CO (*n* = 5) | T2 PD (*n* = 21) |
|---|---|---|---|---|
| L1 SE | 0.036 ± 0.004 | 0.058 ± 0.016 * | 0.036 ± 0.005 | 0.055 ± 0.014 * |
| CI | 2.059 ± 0.076 | 2.631 ± 0.437 * | 10.079 ± 1.061 | 12.074 ± 1.738 * |
| L2 SE | 0.035 ± 0.004 | 0.045 ± 0.009 * | 0.035 ± 0.004 | 0.046 ± 0.010 * |
| CI | 2.046 ± 0.046 | 2.412 ± 0.372 * | 10.219 ± 0.771 | 12.225 ± 1.810 * |
| L3 SE | 0.034 ± 0.004 | 0.045 ± 0.011 * | 0.035 ± 0.004 | 0.046 ± 0.009 * |
| CI | 2.011 ± 0.136 | 2.422 ± 0.396 * | 10.224 ± 0.885 | 12.217 ± 1.812 * |
| L4 SE | 0.053 ± 0.003 | 0.063 ± 0.013 | 0.055 ± 0.004 | 0.065 ± 0.011 |
| CI | 2.753 ± 0.228 | 3.161 ± 0.442 | 13.703 ± 0.714 | 15.326 ± 1.493 * |
| L5 SE | 0.072 ± 0.024 | 0.071 ± 0.023 | 0.075 ± 0.026 | 0.072 ± 0.015 |
| CI | 2.989 ± 0.428 | 3.416 ± 0.726 | 14.024 ± 1.191 | 15.435 ± 1.867 |
| L6 SE | 0.052 ± 0.006 | 0.064 ± 0.012 * | 0.054 ± 0.005 | 0.072 ± 0.015 * |
| CI | 2.644 ± 0.233 | 3.350 ± 0.596 * | 12.555 ± 1.789 | 14.813 ± 1.799 * |
| L7 SE | 0.054 ± 0.007 | 0.066 ± 0.016 | 0.057 ± 0.007 | 0.072 ± 0.013 * |
| CI | 2.624 ± 0.097 | 3.372 ± 0.506 * | 11.981 ± 1.453 | 15.296 ± 1.596 * |
| L8 SE | 0.057 ± 0.014 | 0.064 ± 0.020 | 0.060 ± 0.015 | 0.074 ± 0.020 |
| CI | 2.888 ± 0.366 | 3.508 ± 0.674 | 12.460 ± 1.035 | 14.592 ± 1.512 * |
| Total force L SE | 0.099 ± 0.008 | 0.104 ± 0.017 | 0.103 ± 0.008 | 0.111 ± 0.016 |
| CI | 4.008 ± 0.245 | 4.312 ± 0.582 | 16.008 ± 0.382 | 15.096 ± 1.215 |
| R1 SE | 0.034 ± 0.003 | 0.051 + 0.011 * | 0.036 ± 0.004 | 0.052 ± 0.011 |
| CI | 2.031 ± 0.144 | 2.509 ± 0.402 * | 10.152 ± 0.831 | 11.654 ± 1.659 |
| R2 SE | 0.034 ± 0.002 | 0.047 ± 0.009 * | 0.034 ± 0.002 | 0.166 ± 0.544 |
| CI | 1.973 ± 0.086 | 2.420 ± 0.355 * | 10.079 ± 0.724 | 12.095 ± 1.832 * |
| R3 SE | 0.036 ± 0.004 | 0.047 ± 0.011 * | 0.036 ± 0.004 | 0.045 ± 0.008 * |
| CI | 1.994 ± 0.124 | 2.416 ± 0.347 * | 10.218 ± 0.916 | 12.006 ± 1.908 |
| R4 SE | 0.058 ± 0.015 | 0.061 ± 0.012 | 0.055 ± 0.007 | 0.064 + 0.009 |
| CI | 2.819 ± 0.260 | 3.080 ± 0.429 | 14.114 ± 1.039 | 15.083 ± 1.889 |
| R5 SE | 0.065 ± 0.011 | 0.063 ± 0.012 | 0.060 ± 0.007 | 0.072 + 0.011 * |
| CI | 2.963 ± 0.235 | 3.258 ± 0.571 | 14.363 ± 1.020 | 15.185 ± 1.869 |
| R6 SE | 0.053 ± 0.001 | 0.063 ± 0.009 * | 0.056 ± 0.003 | 0.069 ± 0.009 * |
| CI | 2.683 ± 0.293 | 3.258 ± 0.446 * | 12.542 ± 2.054 | 14.839 ± 1.742 * |
| R7 SE | 0.052 ± 0.006 | 0.061 ± 0.009 * | 0.052 ± 0.004 | 0.070 ± 0.010 * |
| CI | 2.535 ± 0.372 | 3.279 ± 0.530 * | 11.809 ± 1.859 | 15.293 ± 2.137 * |
| R8 SE | 0.052 ± 0.001 | 0.065 ± 0.012 * | 0.054 ± 0.002 | 0.070 ± 0.011 * |
| CI | 2.855 ± 0.221 | 3.486 ± 0.449 * | 12.656 ± 1.674 | 14.919 ± 1.762 * |
| Total force R SE | 0.100 ± 0.010 | 0.104 ± 0.018 | 0.104 ± 0.008 | 0.110 ± 0.016 |
| MSE | 4.042 ± 0.240 | 4.429 ± 0.593 | 16.102 ± 0.456 | 15.374 ± 1.084 |

Data are expressed as mean + SD. CO = healthy controls; PD = Parkinson's disease; * $p < 0.05$. Significant difference between PD and CO.

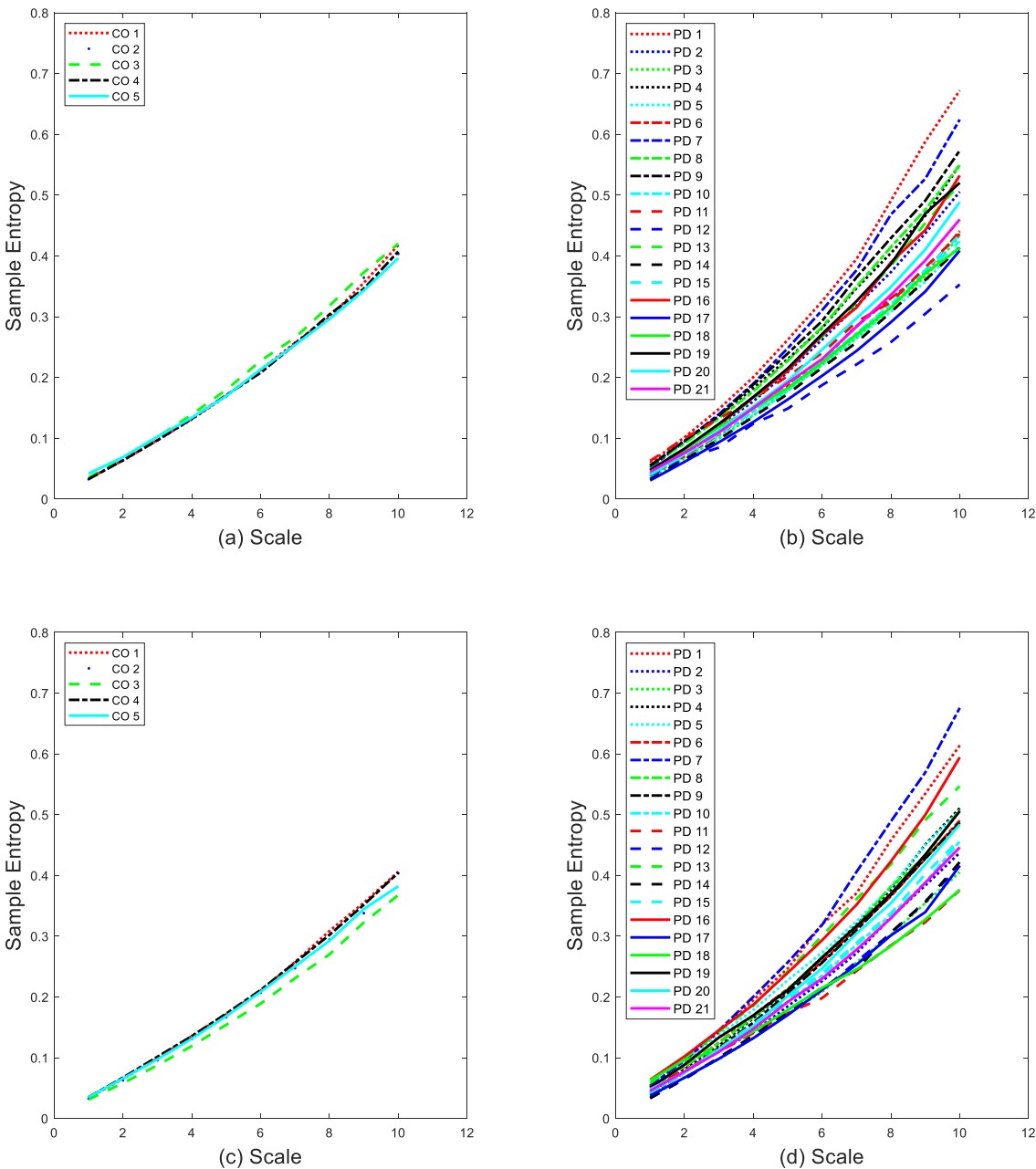

**Figure 4.** T1 MSE of CO and PD (**a**) CO T1: L2; (**b**) PD T1: L2; (**c**) CO T1: R2; (**d**) PD T1: R2.

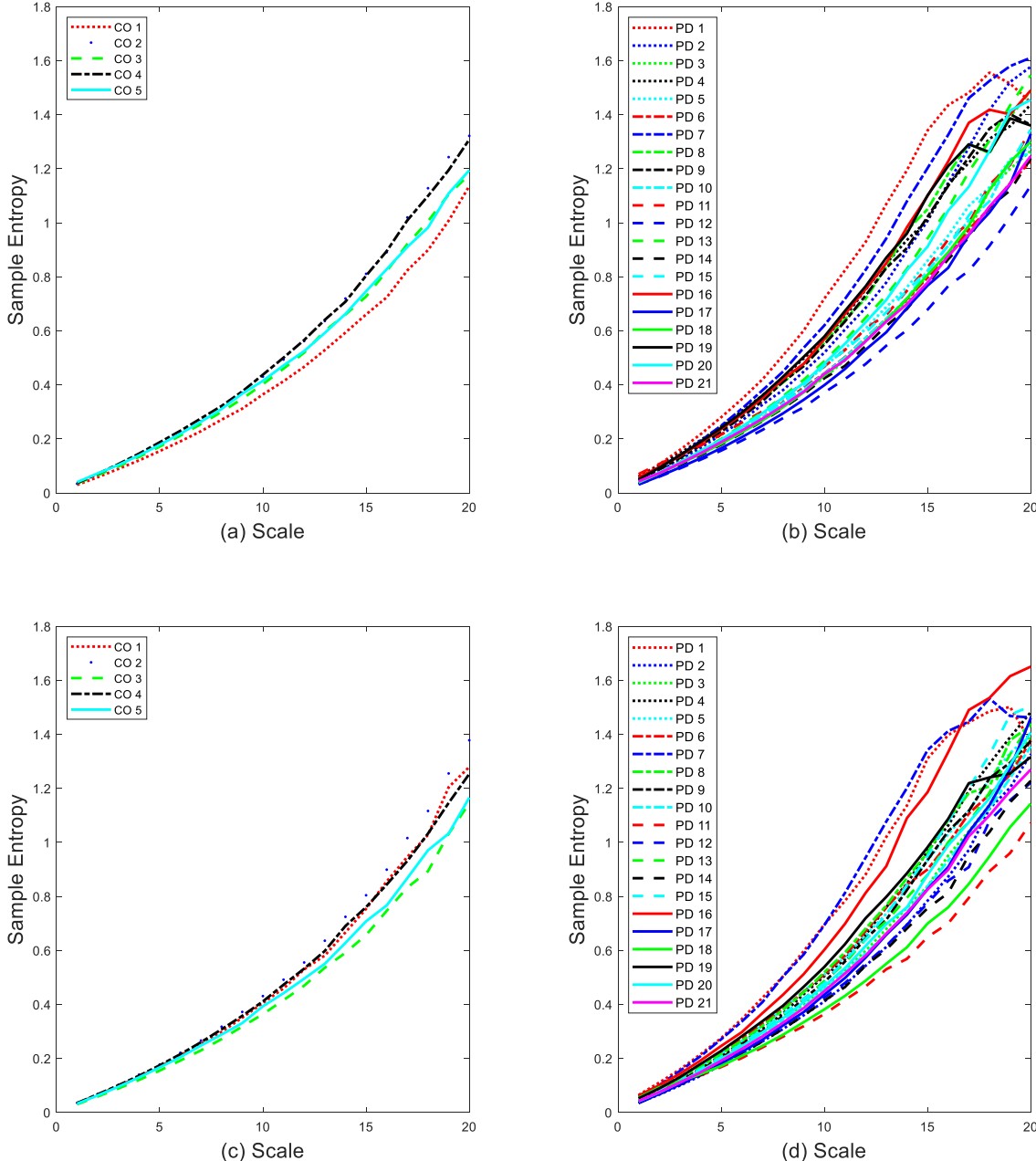

**Figure 5.** T2 MSE of CO and PD (**a**) CO T2: L2; (**b**) PD T2: L2; (**c**) CO T2: R2; (**d**) PD T2: R2.

### 3.2. Comparison of COw and COd CIs

The CO data were analyzed in order to show differences when healthy people performed dual tasks, as shown in Table 3, with one case shown in Figure 7. From Figure 7a,b, it is clear that some of the COd curves values were higher, which can be the reason that only this example of R3 showed significant differences. On the other hand, there were no differences within two groups in Figure 7c,d as a result of the large value differences in COd. The T1 results showed that a right sensor and total ground reaction force of the right foot had *p* value less than 0.05, but there was no significant difference in T2. In addition, most of dual-task data were larger, and this may caused by more time spent adjusting the gait leading to higher gait complexity when doing serial 7 subtraction. It was also noticed that after 20 s, the dual-task data were not different from those of the COw group, and this may indicate that the adjustment was completed, and so these were not statistically different for T2 data.

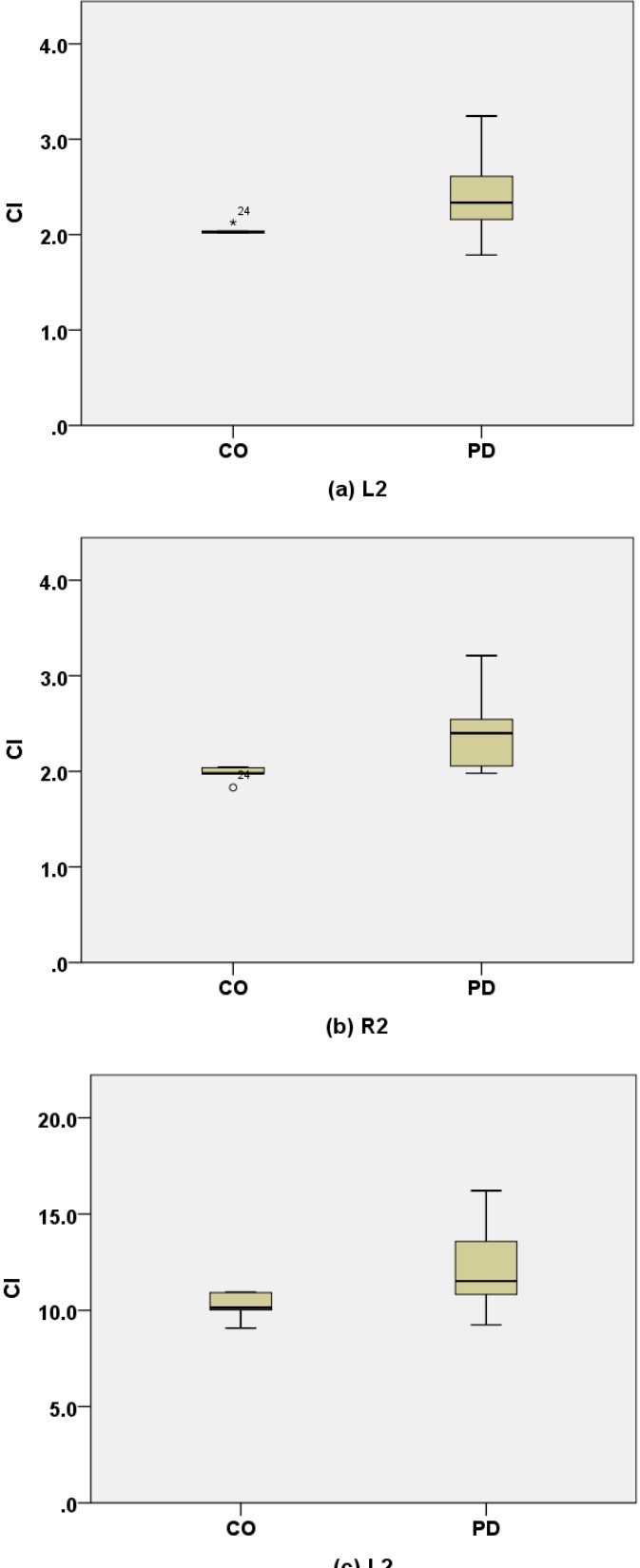

**Figure 6.** *Cont.*

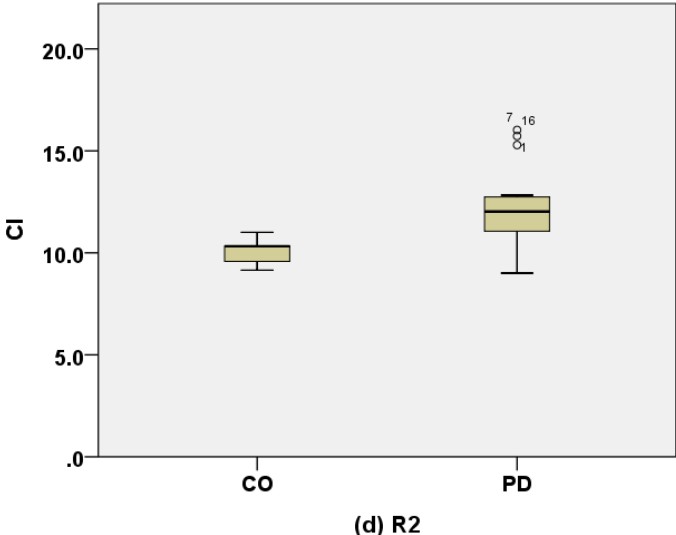

**(d) R2**

**Figure 6.** Boxplot of T1 and T2 CI, CO and PD (**a**) T1: L2; (**b**) T1: R2; (**c**) T2: L2; (**d**) T2: R2. (1, 7, 16: PD subjects; 24: the third of the CO subjects; °: outliners; *: extreme outliners—values that overpass three times the height of the box).

**Table 3.** CI of T1 and T2 for COw and COd.

|  | T1 COw (*n* = 5) | T1 COd (*n* = 5) | T2 COw (*n* = 5) | T2 COd (*n* = 5) |
|---|---|---|---|---|
| L1 | 2.059 ± 0.076 | 2.012 ± 0.262 | 10.079 ± 1.061 | 8.830 ± 2.361 |
| L2 | 2.046 ± 0.046 | 2.101 ± 0.166 | 10.219 ± 0.771 | 9.205 ± 1.247 |
| L3 | 2.011 ± 0.136 | 2.062 ± 0.173 | 10.224 ± 0.885 | 9.293 ± 1.044 |
| L4 | 2.753 ± 0.228 | 2.747 ± 0.158 | 13.703 ± 0.714 | 12.955 ± 1.827 |
| L5 | 2.989 ± 0.428 | 2.942 ± 0.382 | 14.024 ± 1.191 | 13.495 ± 1.580 |
| L6 | 2.644 ± 0.233 | 2.658 ± 0.217 | 12.555 ± 1.789 | 11.920 ± 1.900 |
| L7 | 2.624 ± 0.097 | 2.681 ± 0.204 | 11.981 ± 1.453 | 11.593 ± 1.131 |
| L8 | 2.888 ± 0.366 | 2.894 ± 0.373 | 12.460 ± 1.035 | 12.402 ± 1.764 |
| Total force L | 4.008 ± 0.245 | 4.068 ± 0.150 | 16.008 ± 0.382 | 15.140 ± 1.197 |
| R1 | 2.031 ± 0.144 | 2.099 ± 0.206 | 10.152 ± 0.831 | 10.012 ± 1.531 |
| R2 | 1.973 ± 0.086 | 2.081 ± 0.142 | 10.079 ± 0.724 | 9.897 ± 1.425 |
| R3 | 1.994 ± 0.124 | 2.100 ± 0.162 * | 10.218 ± 0.916 | 9.937 ± 1.565 |
| R4 | 2.819 ± 0.260 | 2.816 ± 0.252 | 14.114 ± 1.039 | 13.475 ± 1.954 |
| R5 | 2.963 ± 0.235 | 2.988 ± 0.239 | 14.363 ± 1.020 | 13.889 ± 1.125 |
| R6 | 2.683 ± 0.293 | 2.698 ± 0.168 | 12.542 ± 2.054 | 11.959 ± 1.310 |
| R7 | 2.535 ± 0.372 | 2.548 ± 0.185 | 11.809 ± 1.859 | 11.496 ± 0.474 |
| R8 | 2.855 ± 0.221 | 2.909 ± 0.178 | 12.656 ± 1.674 | 12.3370 ± 1.762 |
| Total force R | 4.042 ± 0.240 | 4.170 ± 0.293 * | 16.102 ± 0.456 | 14.602 ± 2.119 |

Data are expressed as mean ± SD. CO = healthy controls; * $p < 0.05$ between COw and COd.

### 3.3. Comparison of CI between PDw and PDd

Apart from the comparison of the two groups from CO, signals from L4, R5, and right total ground reaction force data of PD group in T2 showed significant difference, with the dual-task group having lower values. Studies indicated that PD patients had impaired cognitive functions, so the changes and adaptation of the gait happened slower than in the CO group, which is seen in T2. The data are shown in Table 4 and Figure 8. Figure 8a,b shows that the curves were identical, hence no significant differences might be a possible explanation. However, when looking at Figure 8b,d, PD dual seems to have diverse values, but mostly the curves from PD posed a higher position than their counterparts.

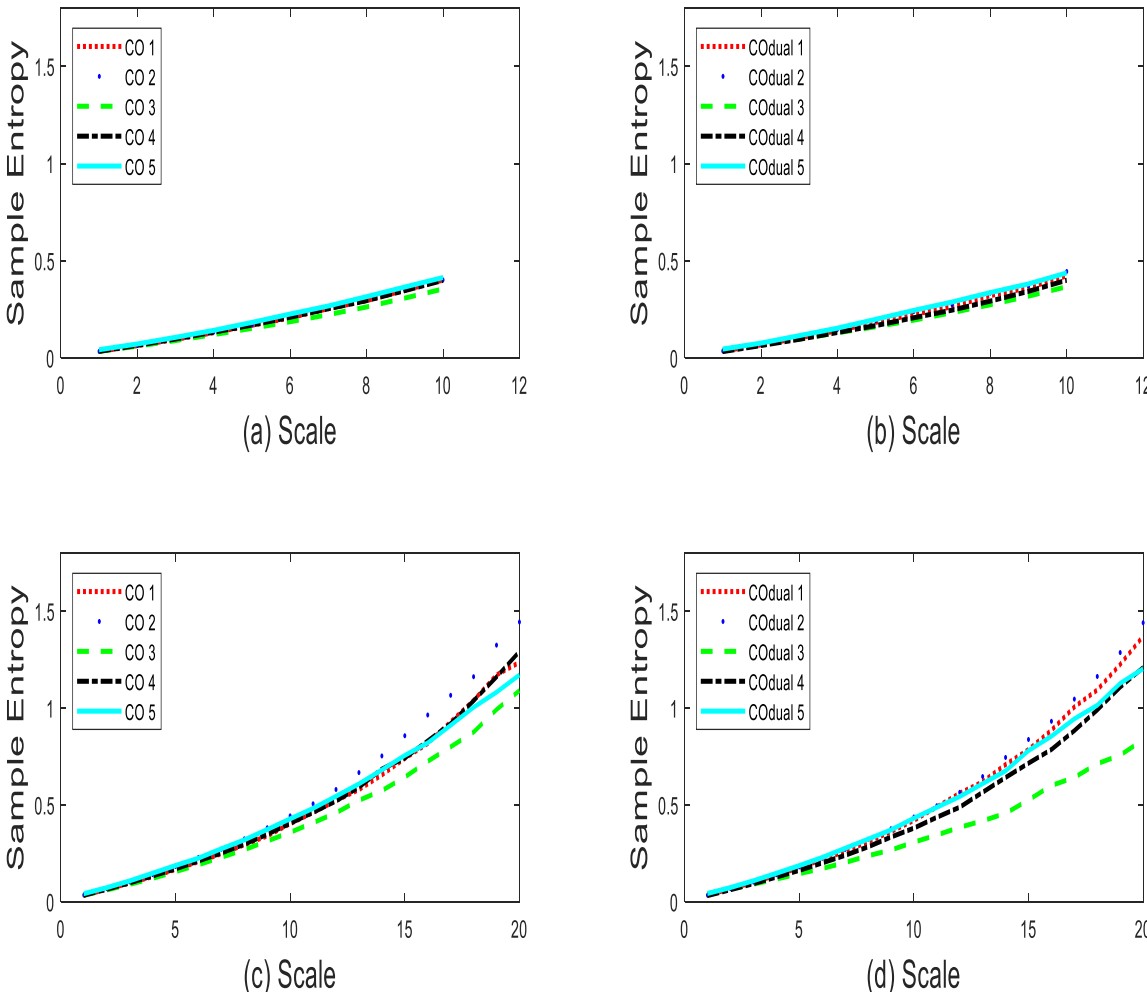

**Figure 7.** T1 and T2 R3 MSE of COw and COd (**a**) COw: T1; (**b**) COd: T1; (**c**) COw: T2; (**d**) COd: T2.

**Table 4.** CI of T1 and T2 for PDw and PDd.

|  | **T1 PDw (*n* = 21)** | **T1 PDd (*n* = 21)** | **T2 PDw (*n* = 21)** | **T2 PDd (*n* = 21)** |
|---|---|---|---|---|
| L1 | 2.631 ± 0.437 | 2.656 ± 0.483 | 12.074 ± 1.738 | 12.125 ± 1.767 |
| L2 | 2.412 ± 0.372 | 2.479 ± 0.411 | 12.225 ± 1.810 | 12.189 ± 1.961 |
| L3 | 2.422 ± 0.396 | 2.452 ± 0.378 | 12.217 ± 1.812 | 12.172 ± 1.930 |
| L4 | 3.161 ± 0.442 | 3.105 ± 0.475 | 15.326 ± 1.493 | 14.787 ± 1.443 * |
| L5 | 3.416 ± 0.726 | 3.302 ± 0.566 | 15.435 ± 1.867 | 15.175 ± 1.720 |
| L6 | 3.350 ± 0.596 | 3.298 ± 0.607 | 14.813 ± 1.799 | 14.515 ± 1.957 |
| L7 | 3.372 ± 0.506 | 3.362 ± 0.549 | 15.296 ± 1.596 | 15.154 ± 1.829 |
| L8 | 3.508 ± 0.674 | 3.670 ± 0.658 | 14.592 ± 1.512 | 14.895 ± 1.716 |
| Total force L | 4.208 ± 0.614 | 4.352 ± 0.608 | 15.022 ± 1.205 | 15.074 ± 0.993 |
| R1 | 2.509 ± 0.402 | 2.533 ± 0.541 | 11.654 ± 1.659 | 11.339 ± 2.158 |
| R2 | 2.420 ± 0.355 | 2.428 ± 0.471 | 12.095 ± 1.832 | 11.769 ± 2.255 |
| R3 | 2.416 ± 0.347 | 2.426 ± 0.469 | 12.006 ± 1.908 | 11.845 ± 2.440 |
| R4 | 3.080 ± 0.429 | 3.056 ± 0.498 | 15.083 ± 1.889 | 14.708 ± 2.126 |
| R5 | 3.258 ± 0.571 | 3.246 ± 0.616 | 15.185 ± 1.869 | 14.700 ± 2.222 * |
| R6 | 3.258 ± 0.446 | 3.245 ± 0.521 | 14.839 ± 1.742 | 14.480 ± 2.078 |
| R7 | 3.279 ± 0.530 | 3.358 ± 0.633 | 15.293 ± 2.137 | 14.933 ± 2.441 |
| R8 | 3.486 ± 0.449 | 3.606 ± 0.584 | 14.919 ± 1.762 | 14.832 ± 2.080 |
| Total force R | 4.429 ± 0.593 | 4.348 ± 0.550 | 15.374 ± 1.084 | 15.014 ± 1.191 * |

* *p* < 0.05 between PDw and PDd.

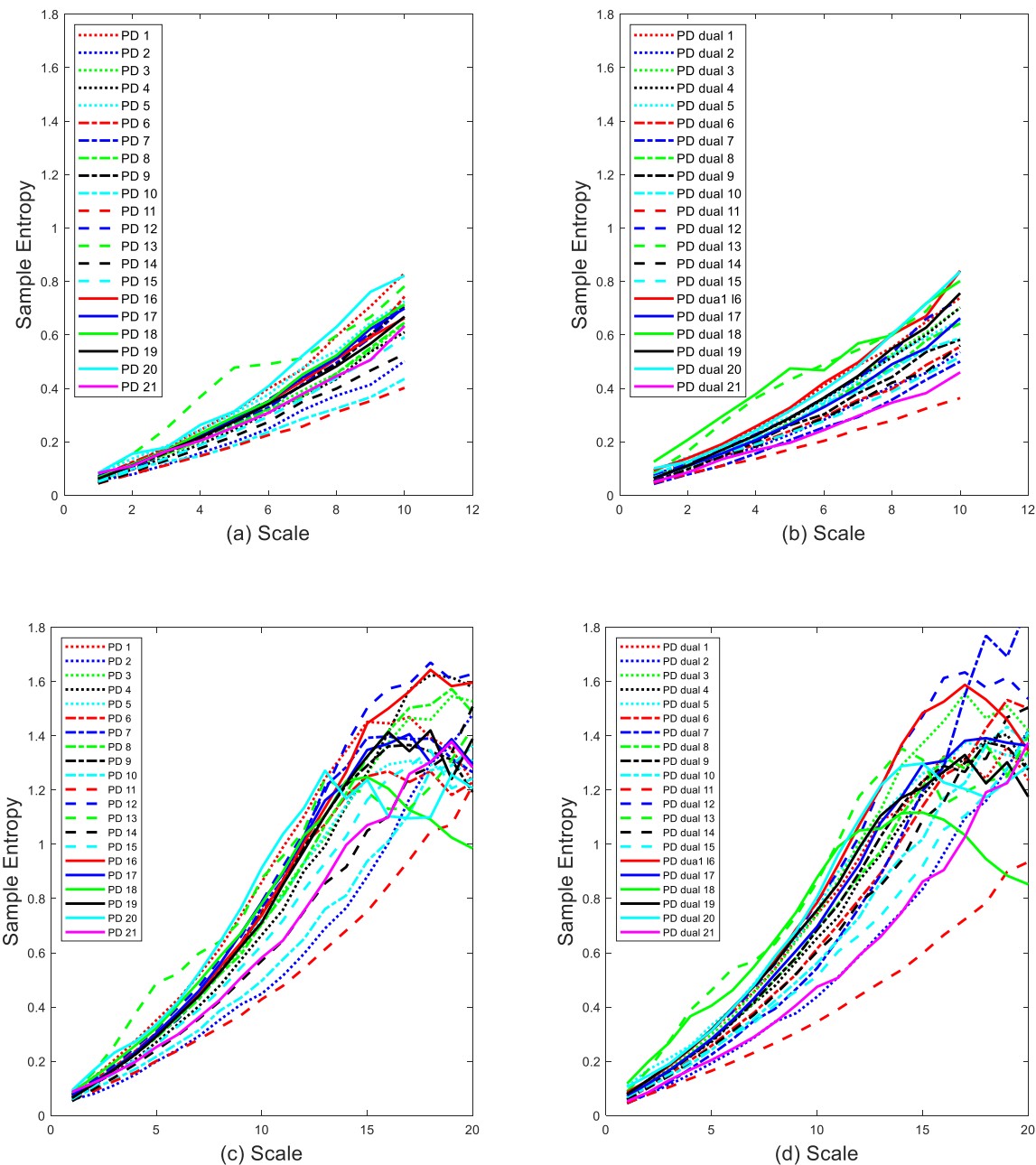

**Figure 8.** T1 and T2 R5 MSE of PDw and PDd (**a**) PDw: T1; (**b**) PDd: T1; (**c**) PDw: T2; (**d**) PDd: T2.

*3.4. Comparison of CO and PD Mean CI*

The total ground reaction force was measured by eight sensors for each foot. The data are shown in Table 5 and the boxplot in Figure 9. As the CI data indicate, the mean of the left and right ground reaction forces (16 data points) of the different groups showed a significant difference ($p$ = 0.002, 0.001), with the PD group having larger values. For the SE data, there were significant differences ($p$ = 0.013) between groups.

**Table 5.** Mean of SE and CI of T1 and T2 for CO and PD.

|  | CO T1 | PD T1 | CO T2 | PD T2 |
|---|---|---|---|---|
| SE Mean $\pm$ SD | 0.049 $\pm$ 0.012 | 0.058 $\pm$ 0.008 * | 0.049 $\pm$ 0.123 | 0.069 $\pm$ 0.028 * |
| Mean $\pm$ SD | 2.492 $\pm$ 0.397 | 2.999 $\pm$ 0.440 * | 11.949 $\pm$ 1.604 | 13.941 $\pm$ 1.536 * |

* $p < 0.05$ between PD and CO.

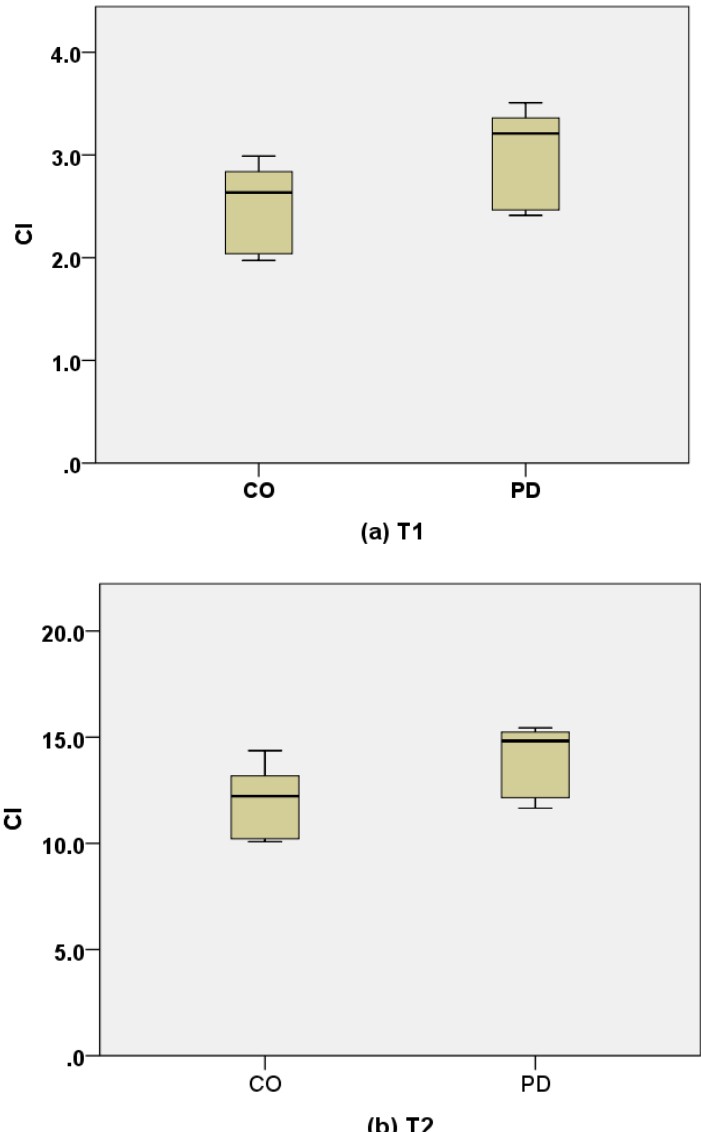

**Figure 9.** Boxplot of T1 and T2 CI of CO and PD (**a**) T1 and (**b**) T2.

### 3.5. Comparison of COw and COd Mean CI

Comparisons between the mean MSE of COw and COd are shown in Table 6 and Figure 10. The results showed that the mean of the left and right ground reaction forces of the different groups had significant differences ($p$ = 0.021, 0.000), with the COd group having larger values in T1 but lower values in T2. This meant that when people deal with stress, MSE is increased first and then decreases when the stress has overwhelmed them.

**Table 6.** Mean of CI of T1 and 2 for COw and COd.

|  | COw T1 | COd T1 | COw T2 | COd T2 |
|---|---|---|---|---|
| Mean ± SD | 2.492 ± 0.397 | 2.521 ± 0.374 * | 11.949 ± 1.604 | 11.418 ± 1.673 * |

\* $p < 0.05$ between COw and COd.

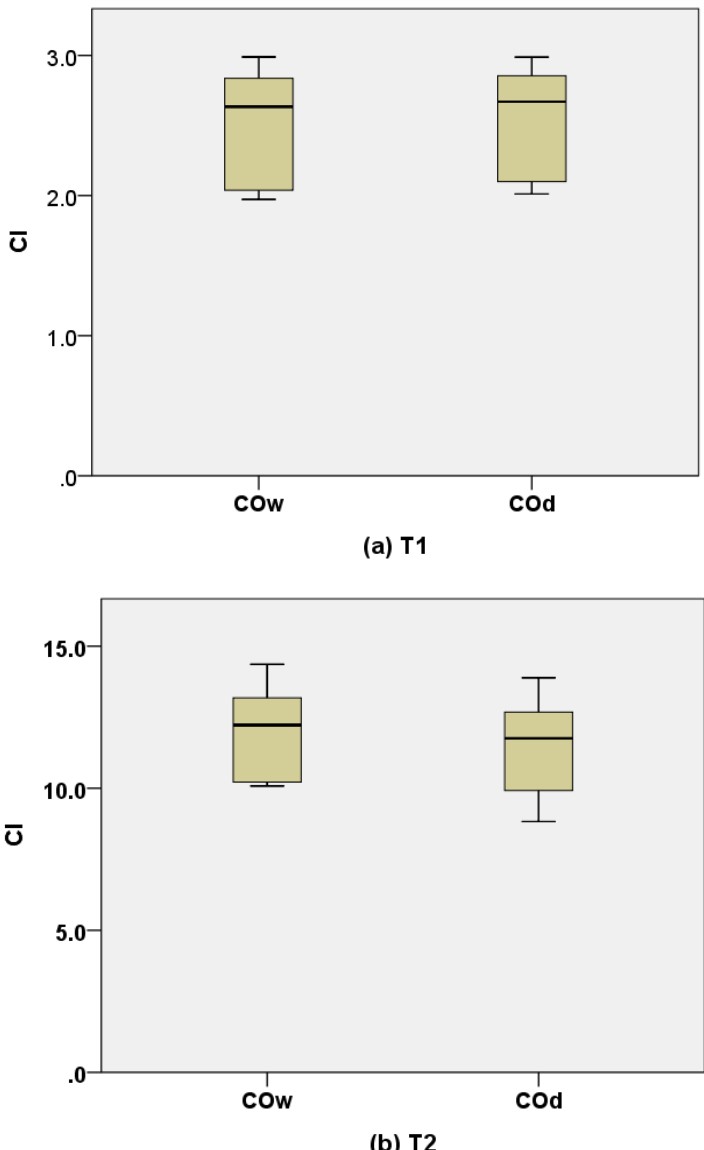

**Figure 10.** Boxplot of T1 and T2 CI of COw and COd (**a**) T1 and (**b**) T2.

### 3.6. Comparison of PDw and PDd Mean CI

Last, the difference between the CI means of the PDw and PDd tasks was compared, the data are shown in Table 7 and Figure 11. The results showed that the mean of the left and right ground reaction force between groups had a significant difference ($p = 0.001$) in T2, with the PDd group having lower values. For PD patients, the disease itself is a stress causing CI to increase. When PD patients deal with dual tasks, more stress is applied and as a result CI is decreased further.

**Table 7.** Mean of CI of T1 and 2 for PDw and PDd.

|  | PDw T1 | PDd T1 | PDw T2 | PDd T2 |
| --- | --- | --- | --- | --- |
| Mean ± SD | 2.999 ± 0.440 | 3.014 ± 0.444 | 13.941 ± 1.536 | 13.726 ± 1.479 * |

* $p < 0.05$ between PDw and PDd.

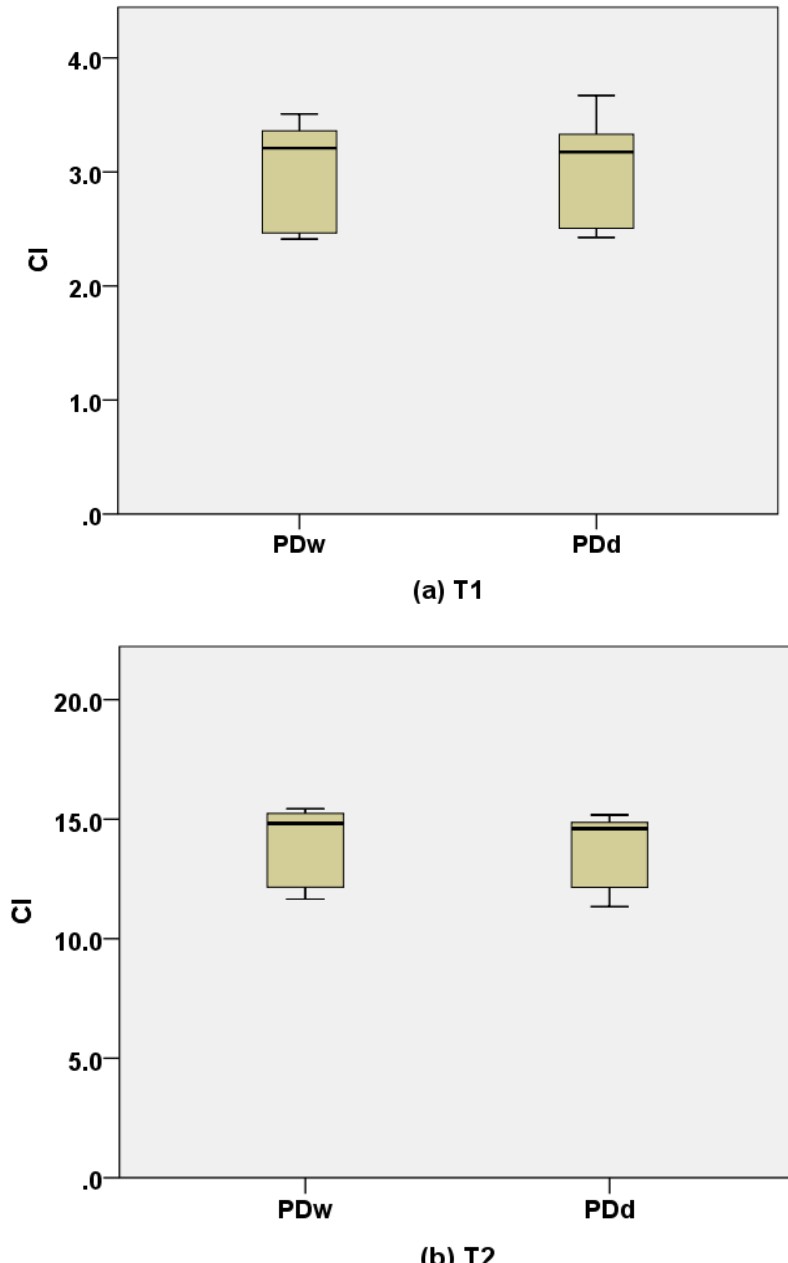

**Figure 11.** Boxplot of T1 and T2 CI of PDw and PDd (**a**) T1 and (**b**) T2.

## 4. Discussion

From the demographic data, it can be seen that there were significant differences between groups in age, weight, BMI, and gait speed. Due to the availability of the recruited subjects in this experiment, there were only five people in the CO group, which made it difficult to match similar data. This can be explained since the data were retrieved by open access, so there was no control over the significant differences between demographics. The interesting part is that although the demographics were not perfectly matched, the gait speed of CO was still greater than that of PD patients. This result is compatible with other studies about reduced gait speed in PD patients [2]. In addition, it was also shown that the dual task decreased gait speed in both groups [4]. Unfortunately, there were no significant differences between stride time, and since the database did not provide stride length, it cannot be concluded that shortened stride can be seen in Parkinson's disease patients. It means that traditional gait signals are reliable for some signals to see the difference, but some may not. That is why MSE is used to fight this inconsistency.

When considering parameters used in MSE analysis, one may be concerned about inappropriate filtering. To find an appropriate value for the filtering level ($\gamma$), a suitable standard deviation is required, but in the meantime, an increase of scales may further influence the results of sample entropy. As mentioned in the results, MSE changes when subjects need to take more effort to adapt their gait back to normal. For example, for PD patients and CO with dual tasks in T1, gait complexity (CI) was increased. When it comes to the CO dual task in T2, MSE decreased and that represents that even though they are healthy controls, dual tasks are still seen as stress or noise, so they need time to deal with secondary tasks. However, larger gait complexity did not improve the gait speed.

On the other hand, for PD patients with dual tasks, CI decreased in T2, and the time needed for adaptation was longer than that for CO. The results were not the same as mentioned earlier. The possible reason may be the impaired cognition function of PD and the use of other strategies when coping with dual tasks. Normally, healthy people increased gait complexity at first, but PD patients appeared to have a higher baseline MSE due to their disease; so further increasing complexity to adapt to this stress may not be a possibility for them. They had to seek other ways to try maintaining walking while challenging their mental acuity as well. With more stress on them, their complexity decreased due to not coping with this stress. The results had similar findings with the study of noise-enhanced tactile sensation [32], indicating that when noise increases, people perform better to a peak and then their performance decreases soon after while the noise is still increasing. This may also occur naturally: biological systems have evolved the capability to exploit adaptation to the stress first, so the complexity will increase. However, when more or heavy stresses are added on the biological system, it cannot digest all these stresses, so the complexity is then reduced. That is why many papers [10,12,13,16,33–35] have always shown loss of complexity or de-complexification during aging and diseases. In this study, we investigated the early stages of disease (e.g., our PD patients), consequently, biological systems tended to increase the complexity to combat the disease or stress for survival. Due to this property, it may be a good indicator to understand the progress of PD.

The strength of this paper is the use of MSE to simplify gait analysis, which usually requires the intervention of health care professionals. With the sensors in shoe insoles and the equations mentioned above, we can get the first-hand data of the complexity of gait from various people. Moreover, by comparing ground reaction force data with the MSE data provided in this study, screening of people with gait impairments or those who would potentially develop PD can be realistic. Therefore, this study can give us a quick insight about how complexity changes and about the relationship of stress on PD patients.

However, the limitations of this study are associated with the data being acquired from PhysioNet with limited cases. The imbalance between the numbers of the two groups was due to the availability of dual task results, as for the CO group, only five complete data could be found. However, given that there were more data in the target PD group, it was possible to select more subjects and to observe differences within the PD group to try to know the complexity differences between diverse stages of PD. Some data were missed in demographics, so the baseline results may not present the real images of the patients. The number of subjects was not enough to form a complete picture of PD patients. Since the patients were in their early stages, according to the Hoehn–Yahr Scale, most of them may have a weak side, that is the side with more serious symptoms. Unfortunately, the database from PhysioNet did not show these details, and this led to difficulties when analyzing the ground reaction force of the right and left sides. Although gait impairment can be identified, people with Parkinson's have multiple types of gait, and these can be hard to predict merely by gait speed and ground reaction force alone.

When using MSE analysis, there are some limitations that have been noticed. First, the word "complexity" itself is related to the existence of long-range correlations, nonlinear multifractal properties, and/or chaotic dynamics, which are not unequivocally linked to the signal features reflected by entropy measures. In addition, a previous paper [8] mentioned that stationarity of the time series is a prerequisite when applying entropy mea-

sures. Second, different entropy measures lead to diverse results, so it is rather subjective. Third, MSE is usually ignored by researchers due to the problem of filtering the artifacts. Last, some unaddressed issues with the computation of entropy measures via only calculating SD of the original series or coarse-grained calculation by only averaging time series are the effect of long-range correlations. For example, the use of SD of the original series can be modified by multiscale complexity (MSC) analysis, in which each observing scale is evaluated independently via the SamEn using the variance of the coarse-grained time series [36]. It is different from the results of MSE, which uses the same variance of the original signal in a multiscale approach. In their paper, they reported that it performed well to figure out the statistical differences of paired comparisons picked from reactive lymphadenopathy and five categories of lymphomas.

MSE was proposed to characterize complexity as a function of the time-scale factor, $\tau$. However, the use of an averaging filter to calculate each coarse grain will cause poor performances. Many researchers [28,29,37] have proposed different methods to solve this problem. The refinement of the procedure for the elimination of the fast temporal scales is based on the replacement of the FIR filter with a low-pass Butterworth filter called refined multiscale entropy (RMSE) [28]. However, MSE and RMSE lack an analytical framework that allows their calculation for known dynamic processes and cannot be reliably computed over short time series. To overcome these limitations, Faes et al. [29] proposed a method to assess RMSE for autoregressive (AR) stochastic processes called linear MSE (LMSE). Finally, when dealing with the issue of ectopic beats, Lin et al. [37] adopted the intuitive idea that the coarse-grained time sequence will be better reconstructed with the median value rather than mean value over non-overlapping windows.

Additionally, in comparison to other studies with big samples, this study may not be as powerful because there were only five subjects in the CO group, but the $p$ value undoubtedly showed significant differences between the two groups. However, the statistical analysis can still be considered effective. Even though there were few limitations, it was still possible to come up with an image of PD patients performing dual tasks with difficulties they might encounter and to compare them to healthy individuals. By knowing the differences, PD patients can be assisted during the very early onset of the disease.

### 5. Conclusions

In this study, an advanced method, i.e., the MSE, is used to identify gait problems in PD patients with and without dual tasks. The results indicate that when patients try to adapt their gait due to mild pathology or stress, first the complexity increases to a peak, and then it decreases when they cannot cope with this pathology or stress. In the future, more studies are needed to confirm that the MSE and complexity can be applied to screen the possible early stages of PD patients for treatment to delay them from deteriorating further.

**Author Contributions:** Conceptualization Y.-L.H. and M.F.A.; investigation, Y.-L.H. and M.F.A.; methodology, Y.-L.H. and M.F.A.; software, Y.-L.H.; supervision M.F.A.; validation, Y.-L.H. and M.F.A.; visualization, Y.-L.H. and M.F.A.; writing, Y.-L.H. and M.F.A. All authors have read and agreed to the published version of the manuscript.

**Funding:** This research received no external funding.

**Institutional Review Board Statement:** Not applicable.

**Informed Consent Statement:** Not applicable.

**Data Availability Statement:** This study utilizes the publicly available dataset from https://physionet.org, accessed on 18 May 2021.

**Conflicts of Interest:** The authors declare no conflict of interest.

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
