# Peer review of "Gait Analyses of Parkinson’s Disease Patients Using Multiscale Entropy"

_electronics, doi:10.3390/electronics10212604_

Round 1
Reviewer 1 Report
The paper seems not appropriate for the journal “Electronics”, because it does not enter into any detail of the devices and sensors for acquisition of the gait data, and also because these data were not acquired by the Authors but come from a public database. I recommend resubmission to another MDPI journal, e.g. “Entropy” would be much more appropriate.
The rationale of the work should be better explained. It is not clear what is the meaning of comparing the (multiscale) complexity of gait time series in two different phases (“unstable” gait and the following part of the dataset). About the comparison between controls and Parkinson subjects, if the aim of the work is just to differentiate the two groups, this could be likely done using much more simple measures than the multiscale entropy. If it is true that gait analysis is typically done only at the level of visual inspection, simple statistics (mean, variance, autocorrelation, power spectra) should be investigated before moving to entropy measures. If this is not the case (it is more likely that previous reports analyze quantitatively gait data like these coming from a public database), the entropy analyses performed here should be compared with existing measures.
Also the rationale of using the MSE should be clarified. From the paper it does not emerge clearly why one should span several scales (because of better discriminative capability? To focus on physiological mechanisms deployed over different temporal scales?)
Multiscale Entropy in its original formulation is known to be affected by problems related to inappropriate filtering and rescaling parameters [see, e.g., Valencia JF et al. "Refined multiscale entropy: Application to 24-h holter recordings of heart period variability in healthy and aortic stenosis subjects." IEEE Transactions on Biomedical Engineering 56.9 (2009): 2202-2213; Faes L et al. "Efficient computation of multiscale entropy over short biomedical time series based on linear state-space models." Complexity 2017 (2017)]. These aspects should be considered in order to perform appropriate MSE analysis.
There are no elements in the paper which allow to understand whether MSE analysis is properly computed. Nothing is said about stationarity of the time series, sampling frequency of the data, time series length. Exemplary time series should be shown, stationarity test applied, and dependence on the MSE parameters investigated. Entropy measures are known to be affected by artifacts, trends and nonstationarities [see, e.g., Xiong W et al. (2017). Entropy measures, entropy estimators, and their performance in quantifying complex dynamics: Effects of artifacts, nonstationarity, and long-range correlations. Physical Review E, 95(6), 062114], and this should be carefully taken into account prior to any analysis.
Also the statistical analysis may be improved. The two groups are highly mismatched (21 Parkinson patients and only 5 controls), parametric tests are employed apparently without verifying the normality of the distributions, no corrections for multiple comparisons seems to be used.
Data for different sensors are shown, but the differences between them are not investigated in detail.
Author Response
Gait Analyses of Parkinson’s Disease Patients using Multiscale Entropy
Yuan-Lun Hsieh 1 and Maysam F. Abbod 2,*
Reviewer 1:
Comments and Suggestions for Authors
- The paper seems not appropriate for the journal “Electronics”, because it does not enter into any detail of the devices and sensors for acquisition of the gait data, and also because these data were not acquired by the Authors but come from a public database. I recommend resubmission to another MDPI journal, e.g. “Entropy” would be much more appropriate.
Answer: The detail of the sensors is provided in page 3 to 3 lines 99 to 103. In addition, as you mentioned that it may be more appropriate to submit to another journal, I would like to kindly inform you that this time we submit this Electronics journal is under the special issue ”Wearable Electronics for Assessing Human Motor (dis)Abilities”, By Prof. Dr. Fernanda Irrera et al.
https://www.mdpi.com/journal/electronics/special_issues/weahma
Special Issue Information is in the following:
“The study of human postural, gesture, and gait control systems has a great impact in rehabilitation, sports, and medicine, especially for a concrete objective support to the diagnosis and follow-up, related to diseases involving a reduction in balance and motion abilities.
Such a study can be assessed by electronics, which can play a fundamental role in effectively gathering data later processed by smart algorithms. …”.
Hence, we think this special issue is suitable for our study.
- The rationale of the work should be better explained. It is not clear what is the meaning of comparing the (multiscale) complexity of gait time series in two different phases (“unstable” gait and the following part of the dataset). About the comparison between controls and Parkinson subjects, if the aim of the work is just to differentiate the two groups, this could be likely done using much more simple measures than the multiscale entropy. If it is true that gait analysis is typically done only at the level of visual inspection, simple statistics (mean, variance, autocorrelation, power spectra) should be investigated before moving to entropy measures. If this is not the case (it is more likely that previous reports analyze quantitatively gait data like these coming from a public database), the entropy analyses performed here should be compared with existing measures.
Answer: We may not be that clear in this part, the reason we would like to cut into two phases is because we want to compare the differences between the adaptation of CO and PD. Furthermore, adaptation may be important afterwards when the pressure from pathology or dual task overwhelmed them. By knowing the adaptation when subjects start walking, we can somehow predict the patterns when facing obstacles. Please see our revised manuscript in page 3 lines 118-123.
The data of the gait signals of gait speed and stride time plus the standard deviation were provided in the results in Table 1 (pages 4-5). And we added statistics of stride time in Table 1 and page 15 lines 306 to 314 to explain in the discussion part for significant differences for gait speed and no significant differences for stride time. It means traditional gait signals are not reliable for some signals can see the differences but some may be not. That is why we are trying to use MSE to find this inconsistent. The reason for using MSE is replied in the following.
- Also the rationale of using the MSE should be clarified. From the paper it does not emerge clearly why one should span several scales (because of better discriminative capability? To focus on physiological mechanisms deployed over different temporal scales?)
Answer: More details are added on page 2 lines 50 to 77.
- Multiscale Entropy in its original formulation is known to be affected by problems related to inappropriate filtering and rescaling parameters [see, e.g., Valencia JF et al. "Refined multiscale entropy: Application to 24-h holter recordings of heart period variability in healthy and aortic stenosis subjects." IEEE Transactions on Biomedical Engineering 56.9 (2009): 2202-2213; Faes L et al. "Efficient computation of multiscale entropy over short biomedical time series based on linear state-space models." Complexity 2017 (2017)]. These aspects should be considered in order to perform appropriate MSE analysis.
Answer: Thank you for mentioning about the filtering problem when using MSE analysis. We added on page 4 lines 147 to 152 and page 15 lines 315 to 318 to address these inappropriate filtering and rescaling parameters.
- There are no elements in the paper which allow to understand whether MSE analysis is properly computed. Nothing is said about stationarity of the time series, sampling frequency of the data, time series length. Exemplary time series should be shown, stationarity test applied, and dependence on the MSE parameters investigated. Entropy measures are known to be affected by artifacts, trends and nonstationarities [see, e.g., Xiong W et al. (2017). Entropy measures, entropy estimators, and their performance in quantifying complex dynamics: Effects of artifacts, nonstationarity, and long-range correlations. Physical Review E, 95(6), 062114], and this should be carefully taken into account prior to any analysis.
Answer: We have re-written how to calculate MSE from sample entropy on pages 3-4 lines 130-161. We also address a limitation for calculating entropy on page 16 lines 360-370.
- Also the statistical analysis may be improved. The two groups are highly mismatched (21 Parkinson patients and only 5 controls), parametric tests are employed apparently without verifying the normality of the distributions, no corrections for multiple comparisons seems to be used.
Answer: The explanation was added on page 4 lines 164 to 166. For the normality of the distributions, Levene's Test of Equal Variances is applied first to make sure two groups of variances are equal or not. Then, following the results of the variances, we get two different p values. As a result, the p values we provided went through correction of variances.
- Data for different sensors are shown, but the differences between them are not investigated in detail.
Answer: We added a Figure 2 in page 5 lines 180 to 188. To further interpret the details, an example of the MSE data on each sensor was provided. We can see from the graph and the MSE values, there are subtle differences between each sensor. And even with the same person, both sides demonstrate different characteristics.
Submission Date
12 August 2021
Date of this review
04 Sep 2021 12:22:29
Reviewer 2 Report
The article presents the quantitative measurement analysis using multiscale entropy to identify gait problems in Parkinson’s disease patients with and without dual tasks. Although many articles have comprehensively been reported to analyze gait signals (Entropy 2019, 21(10), 934), Chaos 2016, 26, 023115), I recommend the authors to clearly state their novelty. In addition, I have the following concerns before considering the journal for publication.
Comments:
- The introduction section of the manuscript is weak. I recommend the authors investigate the different entropy-based algorithms in detail and indicate why and how the approach followed in this work is superior to existing reports.
- The reviewer could not find any information about the pressure sensor. What kind of pressure sensors with what performance metrics have been used to measure the gait signal, please clarify.
- It would be much interesting to the broader readership if the authors could have provided gait signals and stride rates too.
- Why the quality of figures are so poor? I recommend the authors include figures that are clear to read. And also, the text size in abscissa and ordinate in most of the figures need to be revised.
Author Response
Gait Analyses of Parkinson’s Disease Patients using Multiscale Entropy
Yuan-Lun Hsieh 1 and Maysam F. Abbod 2,*
Reviewer 2
Comments and Suggestions for Authors
The article presents the quantitative measurement analysis using multiscale entropy to identify gait problems in Parkinson’s disease patients with and without dual tasks. Although many articles have comprehensively been reported to analyze gait signals (Entropy 2019, 21(10), 934), Chaos 2016, 26, 023115), I recommend the authors to clearly state their novelty. In addition, I have the following concerns before considering the journal for publication.
Comments:
- The introduction section of the manuscript is weak. I recommend the authors investigate the different entropy-based algorithms in detail and indicate why and how the approach followed in this work is superior to existing reports.
Answer: Extra description is added on page 2 lines 50 to 77. And to amplify the novelty of this study, we describe on page 2 lines 84-85 as follows. To date, although there are studies using MSE, this is the first paper to identify gait impairment of PD patients using MSE and to identify pathological influences.
- The reviewer could not find any information about the pressure sensor. What kind of pressure sensors with what performance metrics have been used to measure the gait signal, please clarify.
Answer: This has been added on page 3 lines 99 to 103.
- It would be much interesting to the broader readership if the authors could have provided gait signals and stride rates too.
Answer: The data of the gait signals of stride time plus the standard deviation was provided in the results in Table 1 (pages 4-5). In addition, we have added statistics of stride time in Table 1 and page 15 lines 306 to 314 to explain in the discussion part for significant differences for gait speed and no significant differences for stride time. It means traditional gait signals are not reliable for some signals can see the differences but some may be not. That is why we are trying to use MSE to find this inconsistent.
- Why the quality of figures are so poor? I recommend the authors include figures that are clear to read. And also, the text size in abscissa and ordinate in most of the figures need to be revised.
Answer: Please see all the graph provided (pages 3, 5-9, 11-12). We have separated CO and PD in order to see them more clear. Also, we use different line types and add the legend in the figures. Finally, we have replaced all the graph provided by MATALB with the help of export function. By upgrading resolution and to export figures, the figures are more clearly to read and as well as the texts in the figures.
Submission Date
12 August 2021
Date of this review
13 Sep 2021 04:06:37
Reviewer 3 Report
Before publication, I suggest improving the paper.
My comments are below:
All photos are of very poor quality and require a larger font in the description.
The number of lines in Figures 2 and 4 is large and there is no description of these curves. Drawings require a different presentation of the results. The description of these results is also insufficient.
The literature review is too meager. There is only 1 publication from the last two years.
There is no justification as to why this work should be published in Electronics.
Author Response
Gait Analyses of Parkinson’s Disease Patients using Multiscale Entropy
Yuan-Lun Hsieh 1 and Maysam F. Abbod 2,*
Reviewer 3:
Comments and Suggestions for Authors
Before publication, I suggest improving the paper.
My comments are below:
- All photos are of very poor quality and require a larger font in the description.
Answer: Thank you for mentioning this problem. Please see graphs provided (pages 3, 5-9, 11-12). We separate the CO and PD in order to see them more clear. Also, we use different line types and add the legend in figures. Finally, we have replaced all the graph with better quality versions. By upgrading resolution and to export figures, the figures are more clearly to read and as well as the texts in the figures.
The number of lines in Figures 2 and 4 is large and there is no description of these curves. Drawings require a different presentation of the results. The description of these results is also insufficient.
Answer: As mentioned in the 1st point, Figures 2 and 4 are separated to CO and PD in order to make it clearer. Also, we have used different line types and add the legend in these figures. In this revised manuscript, Figure 2 has been replaced to Figures 3 and 4 on pages 7-9 and the description of these results is on page 6 lines 200-207. Also, Figure 4 has been replaced by to Figure 6 on page 11 and the description of these results is on page 10 lines 231-235.
- The literature review is too meager. There is only 1 publication from the last two years.
Answer: Thank you for mentioning this problem. We have revised the introduction part on page 2 lines 50 to 77 and added 4 papers from recent years for references 12-15 as follows:
References 12-15:
- Chu YJ, Chang CF, Weng WC, Fan PC, Shieh JS, Lee WT. Electroencephalography complexity in infantile spasms and its association with treatment response. Clinical Neurophysiology 2021,132(2),480-486
- Weng WC, Chang CF, Wong LC, Lin JH, Lee WT, Shieh JS. Altered resting-state EEG complexity in children with Tourette syndrome: A preliminary study. Neuropsychology 2017;31(4):395–402.
- Miskovic V, MacDonald KJ, Rhodes LJ, Cote KA. Changes in EEG multiscale entropy and power‐law frequency scaling during the human sleep cycle, Hum Brain Mapp. 2019; 40(2): 538–551.
- Tiwari A, Albuquerque I, Parent M, Gagnon JF, Lafond D, Tremblay S, Falk TH. Multi-scale heart beat entropy measures for mental workload assessment of ambulant users. Entropy 2019, 21, 783, 1-20.
- There is no justification as to why this work should be published in Electronics.
Answer: Thank you for your advice. Here are our explanations. This paper is submitted to special issue ”Wearable Electronics for Assessing Human Motor (dis)Abilities”, By Prof. Dr. Fernanda Irrera et al.
https://www.mdpi.com/journal/electronics/special_issues/weahma
Special Issue Information is in the following:
The study of human postural, gesture, and gait control systems has a great impact in rehabilitation, sports, and medicine, especially for a concrete objective support to the diagnosis and follow-up, related to diseases involving a reduction in balance and motion abilities.
Such a study can be assessed by electronics, which can play a fundamental role in effectively gathering data later processed by smart algorithms. …”.
Hence, we think this special issue is suitable for our study.
Submission Date
12 August 2021
Date of this review
14 Sep 2021 16:51:51
Reviewer 4 Report
The authors in this paper use the multi scale entropy to discriminate between classes of patients with Parkinson disease. It is found the the multi scale entropy gives an effective differentiation between affected and unaffected persons. the paper explains briefly the datasets the methodology and a succinct statistical analysis. There are a few issues to improve in an otherwise interesting paper:
1) The presentation of results must be improved: the graphs and plots have poor quality and resolution.
2) The mathematical definition of the sample entropy and the MSE needs to be reviewed. The formulas do bot show in complete form and are difficult to follow if not correctly typeset.
3) The CO groups are rather small, so their statistical significance for the analysis is limited. If they are used only for purposes of comparison against PD, this is fine (as the p values confirm), but should be better discussed by the authors.
Author Response
Gait Analyses of Parkinson’s Disease Patients using Multiscale Entropy
Yuan-Lun Hsieh 1 and Maysam F. Abbod 2,*
Reviewer 4:
Comments and Suggestions for Authors
The authors in this paper use the multi scale entropy to discriminate between classes of patients with Parkinson disease. It is found the the multi scale entropy gives an effective differentiation between affected and unaffected persons. the paper explains briefly the datasets the methodology and a succinct statistical analysis. There are a few issues to improve in an otherwise interesting paper:
- The presentation of results must be improved: the graphs and plots have poor quality and resolution.
Answer: Thank you for mentioning this problem. Please see graphs provided on pages 3, 5-9, 11-12. We have separated CO and PD in order to make it clearer. Also, we have used different line types and add the legend in the figures. Finally, we have replaced all the graph provided by Matlab with the help of export function. By upgrading resolution and to export figures, the figures are more clearly to read and as well as the texts in the figures. Hence, Figures 2 and 4 are separated to CO and PD in order to see them clearer. In this revised manuscript, Figure 2 is replaced by Figures 3 and 4 on pages 7-9 and the description of these results is on page 6 lines 200-207. Also, Figure 4 has been replaced by Figure 6 on page 11 and the description of these results is given on page 10 lines 231-235. Finally, Figure 5 has been changed to Figure 7 on page 12 and the description of these results is on page 11 lines 253-257.
- The mathematical definition of the sample entropy and the MSE needs to be reviewed. The formulas do bot show in complete form and are difficult to follow if not correctly typeset.
Answer: Thank you for the advice. We have re-written how to calculate MSE from sample entropy on pages 3-4 lines 130-161.
3) The CO groups are rather small, so their statistical significance for the analysis is limited. If they are used only for purposes of comparison against PD, this is fine (as the p values confirm), but should be better discussed by the authors.
Answer: Thank you for the advice. We have added on page 16 lines 366 to 370. A statements about small samples and how the p value proves that there is significant differences between two groups.
Submission Date
12 August 2021
Date of this review
08 Sep 2021 12:43:36
Round 2
Reviewer 1 Report
The concerns raised in the first revision are still present.
- The reasons of using a sophisticated statistics like the MSE in this work are not clear. A comparison with simpler metrics (e.g. time- and frequency-domain statistics) is not reported. Also, the rationale of going towards a multiscale analysis instead of simply using the Sample Entropy is not explained, and the benefits of this choice do not emerge from the results.
- MSE computation is affected by parameters which are set without justifying them, e.g. in relation to the sampling frequency.
- The stationarity of the time series is not verified, and the analyzed time series are not shown. Therefore it is impossible to understand whether MSE analysis is properly conducted.
- Improved versions of the MSE, accounting for basic inaccuracies of the first development like the use of the SD of the original series instead that of the rescaled series and the use of a lowpass filter with poor properties (i.e. the averaging filter) are mentioned but not employed.
- The statistical analysis is heavily influenced by the big imbalance between the two groups (21 patients and only 5 controls)
- mathematical notations are poor, and there are errors (e.g., eq. 3 does not follow by substituting eq. 2 in eq. 1, and does not represent the MSE – MSE is not a sum of Sample Entropies)
Author Response
Reviewer 1:
The concerns raised in the first revision are still present.
- The reasons of using a sophisticated statistics like the MSE in this work are not clear. A comparison with simpler metrics (e.g. time- and frequency-domain statistics) is not reported. Also, the rationale of going towards a multiscale analysis instead of simply using the Sample Entropy is not explained, and the benefits of this choice do not emerge from the results.
Answer: Thank you for mentioning this problem. We have added on page 2 lines 64 to 70, page 8 lines 247 to 248, lines 251 to 256, and Table 2, and also on page 15 lines 318 to 319, and Table 5.
The calculation and explanation of SE were added to compare with MSE calculation. Tables 2 and 5 are examples of applying both SE and CI with the same data. Since our signals are between stationary and nonstationary, we can’t apply SE analysis only. This can explain why we choose MSE analysis rather than basic entropy measures.
- MSE computation is affected by parameters which are set without justifying them, e.g. in relation to the sampling frequency.
Answer: Thank you for the advice. We have added on page 3 lines 103 to 107, page 4 lines 144 to 157.
We have explained how the parameters are selected and the relation of the sampling rate.
- The stationarity of the time series is not verified, and the analyzed time series are not shown. Therefore it is impossible to understand whether MSE analysis is properly conducted.
Answer: Thank you for the advice. We have added on page 5 lines 185 to 206, and on page 6 of Figure 2.
We have also discussed the stationarity of the time series in Figure 2 and indicated that our ground reaction force data are between stationary and nonstationary.
- Improved versions of the MSE, accounting for basic inaccuracies of the first development like the use of the SD of the original series instead that of the rescaled series and the use of a lowpass filter with poor properties (i.e. the averaging filter) are mentioned but not employed.
Answer: Thank you for the advice. As you mentioned, we did not employ the improved versions of the MSE in this paper. We appreciate this very much indeed and would keep in mind about these methods to be applied in our future works. So, we discuss these important issues in the limitations of our revised manuscript in the discussion section. We added on page 18 lines 418 to 439.
- The statistical analysis is heavily influenced by the big imbalance between the two groups (21 patients and only 5 controls)
Answer: Thank you for the advice. We have added on pages 17-18 lines 399 to 403. The explanation was added in the limitation of this study.
- mathematical notations are poor, and there are errors (e.g., eq. 3 does not follow by substituting eq. 2 in eq. 1, and does not represent the MSE – MSE is not a sum of Sample Entropies)
Answers: Thank you for mentioning this problem. This is an error. We have corrected and added on page 5 line 179 to 183. The explanation was corrected some errors of the equation and MSE, and substitutes MSE value to complexity index (CI).
Finally, regarding to “(x) Extensive editing of English language and style required”, we have replied as follows.
Answer: The whole revised manuscript has carefully been checked and improved greatly by all authors. The corresponding author, Dr Maysam Abbod, who is currently the Reader in Brunel University London, UK, lives in UK over 25 years. He serves Editorial Board Members of several prestigious journals of MDPI publisher, such as Electronics and Sensors journals. He also serves of Board of Editors of Engineering Applications of Artificial Intelligence journal of Elsevier publisher. Dr. Abbod is not only checking the structure of this revised manuscript but also carefully checking English Language and Style in this time.

Reviewer 2 Report
In the revised manuscript, the authors have satisfactorily addressed most of the concerns raised by the reviewer. The manuscript can be accepted for publication in this journal after minor corrections on some of the figures' abscissa and ordinates (e.g., Figure 5, 8, 9, and 10), which are hard to read.
Author Response
Yuan-Lun Hsieh 1 and Maysam F. Abbod 2,*
Reviewer 2:
- In the revised manuscript, the authors have satisfactorily addressed most of the concerns raised by the reviewer. The manuscript can be accepted for publication in this journal after minor corrections on some of the figures' abscissa and ordinates (e.g., Figure 5, 8, 9, and 10), which are hard to read.
Answer: Thank you for your support and the advice. We appreciate very much indeed. We have changed the font size of abscissa and ordinates in Figures 5, 8, 9, and 10. Now, these are in Figures 6, 9, 10, and 11 because we have added another Figure 2 in this revised manuscript.

Reviewer 3 Report
The paper has been improved but figures 1,5,8,9,10 are still very poor quality, so they are unreadable. They should be changed.
Author Response
Reviewer 3:
- The paper has been improved but figures 1,5,8,9,10 are still very poor quality, so they are unreadable. They should be changed.
Answer: Thank you for the advice. We have changed the size of Figure 1 to present more clearly in this revised manuscript at page 3. Figures 5, 8, 9, and 10, we have changed the font size, title and also changed the size of figures to be more easy to read. Now, they are in figures 6, 9, 10, and 11 because we add another Figure 2 in this revised manuscript.

Reviewer 4 Report
The paper has been significantly improved by the authors according to the previous comments. Both the presentation and the technical background have been improved, and the work is extensive. Thus, I would recommend acceptance gf this manuscript.
Author Response
Reviewer 4:
- The paper has been significantly improved by the authors according to the previous comments. Both the presentation and the technical background have been improved, and the work is extensive. Thus, I would recommend acceptance gf this manuscript.
Answer: Thank you for your support. We appreciate very much indeed.

Round 3
Reviewer 3 Report
the article is nearly ready to be published, but still some figures are very poor quality (graphic resolution) Fig 6,9,10,11
Author Response
Thank you for noticing the quality of the figure. The figures have been enlarge to make it clearer. I think the editorial office will also check the quality of the figures at the production stage.